# Cystic Fibrosis Lung Disease Modifiers and Their Relevance in the New Era of Precision Medicine

**DOI:** 10.3390/genes12040562

**Published:** 2021-04-13

**Authors:** Afsoon Sepahzad, Deborah J. Morris-Rosendahl, Jane C. Davies

**Affiliations:** 1Department of Paediatric Respiratory Medicine, Royal Brompton and Harefield Hospitals, London SW3 6NP, UK; a.sepahzad@rbht.nhs.uk; 2Clinical Genetics and Genomics Laboratory, Royal Brompton and Harefield Hospitals, London SW3 6NP, UK; d.morris-rosendahl@rbht.nhs.uk; 3National Heart & Lung Institute, Imperial College London, Emmanuel Kay Building, 1b Manresa Rd, London SW3 6LR, UK

**Keywords:** cystic fibrosis, modifier genes, lung

## Abstract

Our understanding of cystic fibrosis (CF) has grown exponentially since the discovery of the cystic fibrosis transmembrane conductance regulator (*CFTR*) gene in 1989. With evolving genetic and genomic tools, we have come to better understand the role of *CFTR* genotypes in the pathophysiology of the disease. This, in turn, has paved the way for the development of modulator therapies targeted at mutations in the *CFTR*, which are arguably one of the greatest advances in the treatment of CF. These modulator therapies, however, do not target all the mutations in *CFTR* that are seen in patients with CF and, furthermore, a variation in response is seen in patients with the same genotype who are taking modulator therapies. There is growing evidence to support the role of non-CFTR modifiers, both genetic and environmental, in determining the variation seen in CF morbidity and mortality and also in the response to existing therapies. This review focusses on key findings from studies using candidate gene and genome-wide approaches to identify CF modifier genes of lung disease in cystic fibrosis and considers the interaction between modifiers and the response to modulator therapies. As the use of modulator therapies expands and we gain data around outcomes, it will be of great interest to investigate this interaction further. Going forward, it will also be crucial to better understand the relative influence of genomic versus environmental factors. With this understanding, we can truly begin to deliver personalised care by better profiling the likely disease phenotype for each patient and their response to treatment.

## 1. Introduction

Cystic fibrosis (CF) is an autosomal recessive inherited disorder affecting at least 70,000 people worldwide. It is caused by a mutation in the Cystic Fibrosis Transmembrane Conductance Regulator (*CFTR*) gene, which results in defective function of a critical chloride and bicarbonate channel expressed on the apical surface of various epithelial cells. The consequence of CFTR dysfunction is the disruption of normal ion transport across cell surfaces, leading to thick, sticky mucus in the respiratory, reproductive and digestive tracts [1]. In the lungs, this leads to bacterial infection, inflammation [2], airway scarring (bronchiectasis) [3] and early mortality from respiratory failure. Other complications include pancreatic exocrine insufficiency, bowel disease, diabetes and liver disease [4]. Meconium ileus is seen in newborns and distal intestinal obstruction syndrome (DIOS) in older people [5]. CFTR dysfunction in the genitourinary tract leads to congenital bilateral absence of the vas deferens (CBAVD) and male infertility [4]. Initially considered a paediatric disease with poor prognosis, CF is a very different disease today; several countries are now reporting that over 50% of their CF population is over 18 years old [6].

The identification of the *CFTR* gene was instrumental in enabling genetic research of CF at a molecular level, and since its discovery over 2000 variants have been identified (www.genet.sickkids.on.ca/cftr, accessed on 20 February 2021), although not all are equally likely to cause disease. A large systematic effort (Clinical and Functional Translation of CFTR, CFTR2.org) has to date annotated 442 variants, 360 of which have been found to be disease causing, 48 described as “variants of varying clinical consequence (VVCC)”, 23 as non-CF causing and 11 as of unknown significance. *CFTR* mutations are frequently classified according to their molecular pathology and their impact on CFTR protein function (Table 1) [7].

## 2. Non-CFTR Disease Modifiers

People with the same *CFTR* mutations, even siblings, can demonstrate differing disease phenotypes, suggesting that other factors are involved, for example, genetic variation, environment and behaviours. In the early 1990s, evidence emerged that whilst *CFTR* mutation class was a reasonable predictor of pancreatic disease, it was poor for the lung [8]. This prompted the consideration of non-*CFTR* genetic modifiers [9]. The potential role of so-called “modifier genes” has also been demonstrated by twin and sibling studies, with identical twins demonstrating substantially more similarity than other siblings. One such study suggested that >50% of the variation in lung function could be due to modifier genes [10]. Supporting this, the European CF Twin and Sibling study found that individual factors rather than shared factors determined the lung function of siblings and that differing host response was more important for lung disease progression than shared genetics or the environment [11]. Many modifier gene studies have been conducted; here, we focus on the most significant of these and also consider the evidence for the role of non-genetic modifiers (Figure 1). Whether these will remain important and whether new modifiers, perhaps of drug response, will emerge as many people with CF now have access to drugs restoring CFTR function remains to be seen.

## 3. Genetic Modifiers of CF Lung Disease

These can be roughly grouped into genes involved in: (a) tissue repair mechanisms; (b) host defence and inflammation; (c) epithelial surface ion transport and mucus secretion; (d) response to drug treatments. Different approaches have been taken to study these, including candidate gene studies, genome- or exome-wide association studies.

## 4. Candidate Gene Studies

### 4.1. Nitric Oxide Synthase (NOS)

Airway nitric oxide (NO) is thought to be multi-functional, involved in bronchorelaxation, inflammation and tissue repair responses [12]. There are three recognised isoforms of NOS (NOS-1, NOS-2 and NOS-3) expressed in the respiratory tract. In contrast to other inflammatory lung diseases, levels of fractional exhaled airway nitric oxide (FENO) are decreased in CF [13]. An intronic AAT repeated polymorphism in the *NOS1* gene was shown to modulate FENO levels [14] and be associated with a lower risk of colonisation with *P. aeruginosa* and *Aspergillus fumigatus* [15,16]. A later study supporting this also suggested a milder lung disease phenotype [17]. Grasemann et al. [16] did not observe any differences when investigating a polymorphism in the *NOS-3* gene (894G/T)—which affects the resistance of NOS-3 to proteolysis—in FEV_1_ and similarly did not observe any differences in *P. aeruginosa* infection in their overall cohort. However, sub-analysis found females carrying the mutation to have higher FENO, better FEV_1_ and lower risk of pseudomonas.

### 4.2. Glutathione and Glutathione-S-Transferase (GST)

Glutathione protects the lungs from oxidant induced damage. Several small studies have found an association between polymorphisms in glutathione-S-tranferase *(GST)* and lung function. For example, in one study of 146 children with CF, those homozygous for the *GSTM3*B* allele displayed better lung function [18]. The data around GST have, however, been somewhat conflicting. In three studies, a link was found between the null genotype and poor outcome [19,20]; however, in one of the largest studies looking at the effect of *GST* variants on lung function in CF, no association was found for either *GSTM1*, *GSTP1* or *GSTT1* [21].

### 4.3. Transforming Growth Factor β 1 (TGFβ1)

TGFβ1 is a multifunctional cytokine regulating cell differentiation and proliferation. Early studies reported high levels of this growth factor in bronchoalveolar lavage fluid of people with severe CF lung disease [22]. Several large studies including the Gene Modifier study have demonstrated an association between the CC genotype and a more severe disease state [21], the mechanism of which could be secondary to increased inflammation and fibrotic changes [23]. Interestingly, in this study they reported that alleles in the promoter (−509) and first exon (codon 10) of TGFβ1 are associated with poorer lung function, whilst a different study [24] found an association with worse lung function in opposite alleles to those described by Drumm et al. [21] One potential explanation for these opposing findings is the impact of survivor bias [25]. Several modifier gene studies have been performed in CF, although findings are somewhat discrepant [26]. There appears to be an interesting gene/environment interaction with cigarette smoke [27].

### 4.4. Mannose Binding Lectin 2 (MBL2)

MBL is a pattern recognition molecule of the innate immune system which accumulates in the lung during acute inflammation. It binds to several bacteria including *Staphylococcus aureus* and *P. aeruginosa*. Deficiency in MBL-2 is associated with increased susceptibility to infection [28] and *MBL2* variants have been studied fairly extensively in CF; most have demonstrated an association between low MBL-producing genotypes and more severe lung disease. A meta-analysis concluded that genotypes associated with MBL insufficiency were associated with earlier infection with *P. aeruginosa*, poorer lung function in adults and poorer survival [29]. One study investigated the interplay between TGFβ1 and MBL2 in a cohort of 1019 Canadian paediatric CF patients, reporting that high TGFβ1 production enhanced the modulatory effect of MBL2 on the age of first bacterial infection and the rate of decline of lung function [30]. Whilst the majority of studies have evidenced worse lung function with insufficient *MBL2* genotypes, there have been a few studies showing no effect at all [21,31], and one study even found worse lung function in those that were high producers [32].

### 4.5. Cell Surface Receptors

Studies have considered the role of Toll-like receptors. In one study, a minor association of host defence genotype to basic defect phenotype was found [33], but no associations have been found in other studies [34,35]. The CD14 gene encodes a protein found on the cell membrane of white blood cells. Acting as a receptor for lipopolysaccharide, a component of the outer membrane of Gram-negative bacteria such as *P. aeruginosa*, it plays an important role in the immune response. Alexis et al. [36] showed that the presence of a C allele at position -159 was associated with lower levels of CD14 [36]. Martin et al. [37] used robust methodology to demonstrate that low levels of CD14 in bronchoalveolar lavage (BAL) associated with the CC genotype at position −159 were associated with earlier infection with pseudomonas. Opposing this, this genotype was not found to affect lung function in a study of 105 CF patients [37]. In 2010, a study looking at mutations in *Endothelin receptor type A (EDNRA)* among four different cohorts found that the CC genotype in rs5335 was associated with a 5% absolute reduction in lung function [38].

### 4.6. Macrophage Migration Inhibitory Factor (MIF)

MIF is a key pro-inflammatory mediator. Reduced expression of the *MIF* gene has been seen where there are polymorphisms with five CATT at the −794 promoter. In 2005, Plant et al. [39] showed that in the CF group versus controls, those with a 5-CATT repeat allele had a decreased incidence of *P. aeruginosa* colonisation and milder lung function deficit [39]. Another study found the homozygous 5-repeat genotype at *MIF* −794 to be associated with milder disease in F508del CF patients [40].

### 4.7. Cytokines

Interleukin 8 (IL-8) is a chemoattractant involved in neutrophilic recruitment to the CF lung. In 2008, Hillson et al. [41] illustrated that three polymorphisms in this gene were associated with CF disease severity [41], and trends have been seen with bacterial infection [42]. IL-1 has been found in animal models of chronic infection to play a role in the acquisition and maintenance of *P. aeruginosa* infection. Levy et al. [43] genotyped 58 single nucleotide polymorphisms (SNPs) in the *IL-1* gene cluster in 808 CF patients and identified two particular single nucleotide polymorphisms in the *IL1B* gene that were associated with lung disease severity in CF in both their case–control and family-based studies. The study group further describe an association signal with lung disease at *IL1R* among female F508del-CFTR homozygous patients from the University of North Carolina and Case Western Reserve Cohort [43]. A different study found that the informative microsatellite marker within intron 1 of *IL1R* detects a survival advantage for patients with CF and supports the potential role of interleukin 1 receptor (IL1R) in the pathogenesis of CF [44]. Many other cytokines have also been considered with limited significant findings [26]. 

### 4.8. Ion Channels

*SLC9A3* is a Na^+^/H^+^ exchanger which was shown by Dorfman et al. [45] to influence *P. aeruginosa* infection and lung function in children [45]. Members of the solute carrier family have also been identified as relevant to CF disease severity in genome-wide association studies (GWAS), below. The Epithelial Sodium Channel (ENaC) has also attracted significant attention. This is not surprising since abnormal chloride transport underpins the CFTR mediated primary defect and ENaC transports the sodium counterion. At a molecular level, interactions have been demonstrated between ENaC and CFTR [46] and mutations in the ENaC subunits have been shown to cause Liddle’s syndrome (OMIM 177200) and pseudohypoaldosteronism type 1 (OMIM 264350). A study by Stanke et al. [47] identified *SCNN1B* and *SCNN1G* encoding the β- and γ- of ENaC, respectively, and *TNFRSF1A*, which is in close proximity to *SCNN1A,* encoding the α-subunit as clinically relevant modifiers in CF [47]. It was postulated that blocking ENaCs and the prevention of Na hyperabsorption could serve as a potential treatment strategy [48]. Amiloride, which acts as an ENaC channel inhibitor, promoting sodium excretion, and has potassium sparing properties, was trialled initially with some positive effect in reducing mortality rates. Its use was terminated, however, due to the risk of hyperkalaemia [49]. Two ENaC antagonists QBW 276 (ClinicalTrials.gov Identifier: NCT02566044) and BI 443,651 (ClinicalTrials.gov Identifier: NCT02706925) have been investigated and demonstrated promising safety data in phase 1 trials A different inhalable ENaC inhibitor peptide, SPX-101, has undergone phase 2 trials and has shown significant positive results without causing hyperkalaemia [50]. 

### 4.9. Mucins

Mucins are critically important in airway biology [51] and the secreted mucins (e.g., MUC5AC and MUC5B) are critical for mucociliary clearance. An SNP in the promotor region of MUC5B (rs35705950) is to date the strongest risk allele to be associated with pulmonary fibrosis [52], and mucins have also been considered as possible gene modifiers for cystic fibrosis [53]. In one study of 762 patients, a relationship was found between *MUC5AC* variable number tandem repeat (VNTR) number and disease severity [54]. The 6.4 kb VNTR was associated with more severe disease.

### 4.10. Genes Involved in Drug Responses

β 2 adrenergic receptors (ADRB2) are expressed on airway smooth muscles cells and are involved in bronchoconstriction, and the encoding gene is highly polymorphic. Their involvement in CF lung disease severity is not clear-cut. In one study, there was no significant relationship between Arg16Gly and Gln27Glu polymorphisms in *ADRB2* and the response to bronchodilators [55]. However, another study reported that CF patients with the Arg16Gly polymorphism had better spirometry, although Gln27Glu showed no effect [56]. They also found that the bronchodilator response was reduced in the presence of Gly or Glu allele variants, which opposes the findings by Hart et al. [55]. Other studies of glucocorticoid receptors have identified that the Bcll polymorphism was associated with a faster progression of lung disease and that those with the Bcll GG genotype had a more significant decline in FEV1 [57].

## 5. Genome-Wide Associations Studies (GWAS)

Genome-wide association studies (GWAS) generally investigate single nucleotide DNA variants across the whole genome, offering a non-biased approach and the opportunity for genuinely novel discoveries; they are usually much larger than candidate gene studies. There have been three major GWAS which have identified a number of variants found to have associations with differing phenotypes seen in CF [58,59,60]. In Corvol’s meta-analysis, FEV_1_ was used as a marker for lung disease severity; FEV_1_ is a clinically valuable measure of lung function and a known predictor of survival in CF. One disadvantage, however, is the limitation of using FEV_1_ as a marker of disease severity as a comparison across age groups as FEV_1_ declines with age in CF. Despite this, however, there were several promising findings observed.

Five main loci were found to be associated with variation in lung function [58]. The first of these loci were *MUC4* and *MUC20* on chromosome 3q29. *MUC4* encodes the cell surface associated airway cell mucin 4 and *MUC20* the airway cell mucin 20. Together, they play a crucial role in providing an osmotic barrier and contribute to the periciliary brush layer [61]. In CF, there is an abundance of mucins, which consequently causes imbalance in the osmotic barrier. This imbalance, in turn, causes mucociliary stasis and hence increased risk of infection [30]. No further attempts have been made to replicate the findings for mucin genes outside of these single studies. The next locus identified was chromosome 5p15.33, which contains solute carrier family 9A3 (*SLC9A3*). As described previously, *SLC9A3* has been shown to have a relationship with rates of *P. aeruginosa* infection and lung function in children [45]. Interestingly, *SLC9A3* has been found to influence disease severity of the lung [45,62,63] and gut. In a CF mouse model, the disruption of *SLC9A3* has been found to decrease intestinal obstruction [64]. Similarly, *SLC9A3* has been found to be a modifier of neonatal obstruction in patients with CF [65]. These findings suggest pharmacological therapies targeting epithelial function should be considered alongside those targeting *CFTR.* Furthermore, this evidence for pleiotropy in modifier genes is exciting and could encourage the development of new therapies which confer multi-organ advantages.

The third locus was on chromosome Xp22-q23 and contains solute carrier family 6A14 (*SLC6A14*) and angiotensin II receptor type 2 (*AGTR2*). Whilst *AGTR2* has been suggested to play various roles within the lungs, including the regulation of nitric oxide synthase expression and signalling pathways in lung fibrosis, there is limited support for its role in modifying CF disease. There are no further studies specifically investigating *AGTR2*. There has, however, been further investigation of SLC6A14, which is a sodium and chloride dependent neutral and cationic amino acid transporter. *SLC6A14* has been found to show pleiotropism for meconium ileus and for the lungs with one Canadian study demonstrating an association with lung function as well as infection with *Pseudomonas aeruginosa (P. aeruginosa)* [62]. The mechanism of modulation, however, for *SLC6A14* is not clearly defined. The Genotype Tissue Expression Consortium [66] provided expression quantitative trait locus analysis in lung tissue and found that the risk allele for both phenotypes was associated with increased *SLC6A14* mRNA expression. This conflicts with findings from another study that investigated bronchial epithelial cell cultures in patients with CF and ex vivo lungs and tracheal explants from *SLC6A14* knockout mice, which conversely showed increased *P. aeruginosa* in the presence of *SLC6A14* inhibitor [67]. A recent study by Ahmadi et al. [68] found that arginine transport through *SLC6A14* increased F508del-CFTR Cl^−^ efflux in CF airway epithelial cells stimulated with or without a CFTR corrector, lumacaftor [68]. This induced increase in CFTR function was also observed to lead to an increase in the airway surface liquid (ASL) height. They also found that the potentiation of F508del-CFTR channel function in CF cells induced by SLC6A14 arginine uptake occurred via the nitric oxide (NO) signalling pathway. Intriguingly, they suggest that *SLC6A14* activation may be considered as a complementary therapy to CFTR correction and potentiation in CF patients.

The fourth locus identified was chromosome 6p21.3, which contains Human Leukocyte Antigen (HLA) class II. HLA Class II regulates the immune response via antigen presentation to T-Cells. In the non-CF lung, HLA Class II genes have been associated with asthma [69], susceptibility to allergic bronchopulmonary aspergillosis [70] and variation in lung function in both the CF and non-CF lung [71,72]. In a gene expression association study using lymphoblastoid cell lines from 754 F508del homozygous patients, an association was demonstrated between pathways enriched in HLA Class II genes and the age of onset of persistent *P. aeruginosa* infection [73].

The fifth locus to be identified was on chromosome 11p12-p23, which contains ETS homologous factor (EHF) and Apaf-1 interacting protein (*APIP*). In an early combined GWAS and linkage study [59], chr 11 loci were found to be associated with lung disease in CF patients who were homozygous for F508del. This finding has been replicated in a European cohort [74]. APIP is a methionine salvage pathway enzyme which encodes for Apaf-1-interacting protein, which has been shown to prevent apoptosis in the presence of hypoxia [75] and also plays a role in the inflammatory response [76]. The EHF transcription factor has been found to be involved in F508del processing and to play a role in wound repair and control of epithelial tight junctions [77].

## 6. Whole Exome Sequencing Studies

Whole exome sequencing studies have also been used to investigate the role of modifier genes. One of the main drawbacks of this approach is the large sample sizes needed for adequate power. One strategy to address this disadvantage is comparing the whole exome sequences of those with phenotypes at opposite ends of a spectrum. Studies by Edmond et al. [78,79] identified mutations in dynactin subunit 4 (*DCTN4*) and transmembrane channel-like protein 6 (*TMC6*) to be associated with an earlier age of onset of *P. aeruginosa* infections and also with worse FEV_1_ [78,79]. Caveolin 2 (*CAV2*) conversely was found to have a protective effect. Viel et al. [80] investigated the role of *DCTN4* in 2016, but were only able to replicate findings suggesting the risk of *P. aeruginosa* infection in a subgroup of males who had two class 2 mutations in *CFTR* [80].

## 7. Non-Genetic Modifiers of Lung Disease Severity

As is clear from the above, a substantial amount of research has focused on genetic influences, beyond disease-causing *CFTR* mutations, on disease severity. Somewhat less attention has been paid to non-genetic modifiers, but several have been identified. Socioeconomic status (SES), in particular disadvantage and poverty, has been associated with adverse outcomes in multiple studies [81,82,83], some of which highlight childhood as the time of particular risk. This is not purely a function of ease of access to healthcare, as it is observed in populations with nationally funded health systems as well as insurance-based systems. Environmental factors, some of which may be inter-related with SES, have also been shown to impact CF prognosis: exposure to tobacco smoke (including in utero) [84], air pollution [85], climate and proximity to water, the latter being associated with increased risk of pseudomonas infection [86]. Finally, one of the greatest contributors to health outcomes is adherence to prescribed therapies; studies of behaviour have demonstrated that healthcare professionals are poor both at estimating patient adherence [87] and at encouraging it. Adherence to medication regimes has been shown to vary between 35% and 75% [88,89]. Low adherence has been shown to negatively influence health outcomes such as pulmonary exacerbations in CF [90,91,92]. Numerous apps are being developed specifically to address this unmet need [93,94], and certainly more attention needs to be given to understanding the barriers to adherence and identifying strategies to support adherence to treatment. 

## 8. CFTR Modulator Therapies

Treatments for CF have recently undergone a transformational change with the advent of highly effective CFTR modulator therapies (HEMT). These are systemically administered small molecules restoring CFTR function through either the correction of misfolding (e.g., F508del) or the potentiation of function (correctly located proteins such as G551D, the archetypal gating mutation). Most recently, a triple combination of two correctors (elexacaftor, tezacaftor) and a potentiator (ivacaftor) (ETI) has demonstrated ~14% improvement in FEV_1_ in subjects with a only a single F508del allele [95]. This combination has recently been licensed globally for people with one or two such mutations aged 12 or above; although licensed indications differ somewhat regionally, this represents 80–85% of all people with CF. A substantial decrease in pulmonary exacerbations and hospitalisation has been seen alongside nutritional quality of life improvements following the use of modulator therapies. Longer term data from ivacaftor in patients with G551D demonstrate a decrease in the rate of loss of lung function and rates of chronic pseudomonas infection [96]; changes in biomarkers also support a role for these drugs in circulating inflammatory cells [97]. Given the major impacts HEMT seems likely to have on CF disease trajectories, it is interesting to question how relevant the lung disease modifiers discussed above will continue to be. Intriguingly, there are also plausible hypotheses to suggest that certain modifiers may actually modify the therapeutic response to HEMT, with some data already supportive of this. For the remainder of this article, we will explore this area and speculate on fruitful areas for future research.

## 9. Will Disease Modifiers Continue to Be Relevant in Patients Experiencing Restored CFTR Function from CFTR Modulators?

The answer to this question is probably, “It depends on which modulator drug”. Most people consider ivacaftor and ETI as “highly effective” and dual combinations somewhat less so; it may also depend on when the drug is initiated—here, disease stage is likely more important than age, although they are of course related. Very young infants commencing ivacaftor have not yet been followed for long enough that impacts on lung function and infection rates can be determined, but it seems highly likely that slowing (or even preventing) disease progression will be easier to achieve than reversing structural disease if drugs are started at a later stage. Indeed, pancreatic exocrine insufficiency, long considered irreversible postnatally, has a window for ivacaftor treatment in early life [98]. A genetic or environmental modifier conferring an increased risk of pseudomonas infection, for example, may no longer have a detectable impact if that risk has already been decreased by the early use of HEMT. However, people starting modulators, even HEMT, later in their disease course could theoretically experience greater impacts from disease modifiers, both genetic and environmental, than those lucky enough to start early.

## 10. Is There a Role for Modifiers in the Response to Modulator Therapies?

People with CF with the same mutations and similar baseline disease status respond variably to HEMT, the reasons for which are as yet poorly understood. Modifiers of this drug response could be considered in several categories: genes or co-administered drugs influencing drug metabolism; genetic or environmental influences on the ability of cells in the target organ to respond. Although drug–gene interactions in metabolic pathways have not yet been reported for CFTR modulators, this is an area of growing interest in other areas of medicine and personalised treatments [99] and will likely be explored further in CF. In terms of drug–drug interactions, there are already dose adjustments recommended in prescribing guidelines when, for example, azole antifungals, which are strong CYP3A inhibitors, are required.

To date, only one gene has been reported to modify the clinical response to a CFTR modulator. SLC26A9 is poorly understood but is considered most likely to be an anion channel interacting with CFTR. Based on their observation that the SNP rs7512463 modified CF lung disease in those with a cell surface-located mutation (e.g., gating defect) but not F508del homozygous patients, Strug et al. [66] investigated the association of *SLC26A9* polymorphisms with the response to ivacaftor [66]. They reported on two small cohorts: in the discovery sample (*n* = 11), after adjustments for baseline status, each C allele was associated with an almost 10% increase in FEV_1_ response to ivacaftor. In the replication group (*n* = 13), none of whom were CC genotype, the possession of TC rather than TT was associated with a difference of similar magnitude. The group also showed in vitro that F508del homozygous bronchial epithelia did not demonstrate *SLC26A9* polymorphism-related differences in ion transport, but pre-treated with the corrector, lumacaftor, revealed them. The relationship was replicated in a slightly larger study in France [100], although the impact of the C polymorphism was in the opposite direction, being associated with a poorer treatment response. The authors speculate that rs7512463 is a marker of other gene variants with which it is in linkage disequilibrium, rather than being functional itself. Interestingly, SLC26A9 was not a disease modifier in the French population, either generally or when examining only people with cell surface located *CFTR*. Population-based genetic differences could therefore account for this apparent discrepancy, but the finding does raise questions about the generalisability of either study to other groups. Kmit et al. (2019) [101] demonstrated an association between *SLC26A9* rs7512463 status and the degree of restoration of *CFTR* function in F508del homozygous cells exposed to lumacaftor/ivacaftor; here again, the C allele appeared beneficial. Further work will be needed on this drug response modifier and more widely before a pharmacogenetic approach to drug prescribing is able to be considered. 

## 11. Potential Non-Genetic Modifiers of Response to HEMT

Tobacco smoke impairs *CFTR* function and was associated with reduced benefits of the dual drug, tezacaftor/ivacaftor [102]. Adherence to ivacaftor is not well reflected by self-reports and was demonstrably suboptimal in one small study [103]; data are not yet available for triple therapy. Furthermore, adherence to symptom-directed therapies may fall if people with CF feel substantially better once they start HEMT; as the effects of HEMT include the restoration of cell surface hydration, it may be safe for patients well on treatment to stop other muco-active agents, but this needs confirming; fortunately, two randomised clinical trials (SIMPLIFY, ClinicalTrials.gov identifier: NCT04320381 and CF-STORM) will provide some evidence in this area. *P. aeruginosa* has been shown in vitro to reduce the expression and function of CFTR and to limit the correction of F508del CFTR by lumacaftor [104]. For certain infections, most notably fungi and atypical mycobacteria, the long-term requirement for drugs interacting with current HEMT proves a clinical challenge; interruptions or required dose adjustments to the latter may limit potential health benefits of modulators.

## 12. Discussion

Whilst the discovery of the *CFTR* gene transformed our understanding of CF and provided a target for therapeutic options, it did not fully explain disease pathophysiology. *CFTR* genotype alone is not a good predictor of disease severity, especially in the lung, which led to the hypothesis that there are other factors involved, both genetic and non-genetic. As highlighted in this review, there have been several studies supporting the role of modifier genes, but limitations include the heterogeneity of outcome measures used across studies. Some phenotypes are easier to define than others and there may be interaction with non-genetic modifiers. Another potentially confounding factor is survivor effect and the timings at which the different studies were conducted.

There are also challenges reconciling data from studies using different approaches. A candidate gene approach was the initial approach used, which entails applying knowledge of the pathogenesis of disease in CF to select genes for further study. Many of the early studies had a small sample size, no replication and/or did not show consistent findings [26]. Furthermore, a candidate approach restricts findings to pathways with which we are already familiar. There may also be publication bias towards positive studies. Although associated with higher costs, genome-wide association studies have significantly broadened our knowledge of possible modifier genes and pathways involved in CF. Through GWAS, we are able to obtain data on hundreds and thousands of SNPs that may play a significant role in modifying the disease. The evidence generated from GWAS appears to demonstrate the most promising findings compared to the other approaches used, with novel variants found to modulate disease severity and, more excitingly, some variants showing pleiotropy. The main limitation of GWAS, however, is an inability to draw true causal associations, determining the significance of variations identified and translating the findings to the clinical setting [105]. There are also challenges when an identified locus is associated with disease in one ethnic/regional group and not another, as described above [100]. This may imply differences in risk allele frequency between different geographical regions—hence, much larger sample sizes are needed to detect the association. The combination of GWAS findings and data obtained from whole genome sequencing studies of large cohorts, as they become increasingly available, may contribute to the validation of previous findings in the future. Moreover, whole genome analyses are enabling the identification of common DNA variants that contribute to background polygenic risk and, in future, may allow more accurate prediction of the effect of modifiers on individuals who inherit a monogenic risk, such as in CF. The first polygenic risk scores associated with CF-related diabetes have recently been published [106].

## 13. Conclusions

Our ability to substantially restore the function of *CFTR* with small molecule drugs will likely transform the future of CF for many people, particularly those who commence treatment early in their disease course. Whether the disease modifiers identified in untreated populations will continue to be relevant in those receiving HEMT remains to be seen. As the response to these therapies is variable, there will likely be more interest in factors influencing this in the future. To date, the only genetic modifier reported is SLC26A9, a poorly understood solute carrier thought to interact with cell surface *CFTR* [101]. Two studies reporting associations as well as laboratory data support the association, although population differences mean that this effect is currently not consistent and further work is needed. Meanwhile, environmental/behavioural influences should not be overlooked and may be more amenable to intervention: suboptimal treatment adherence (to either the HEMT or symptom-targeted treatments) and exposure to cigarette smoke being two issues meriting particular attention.

## Figures and Tables

**Figure 1 genes-12-00562-f001:**
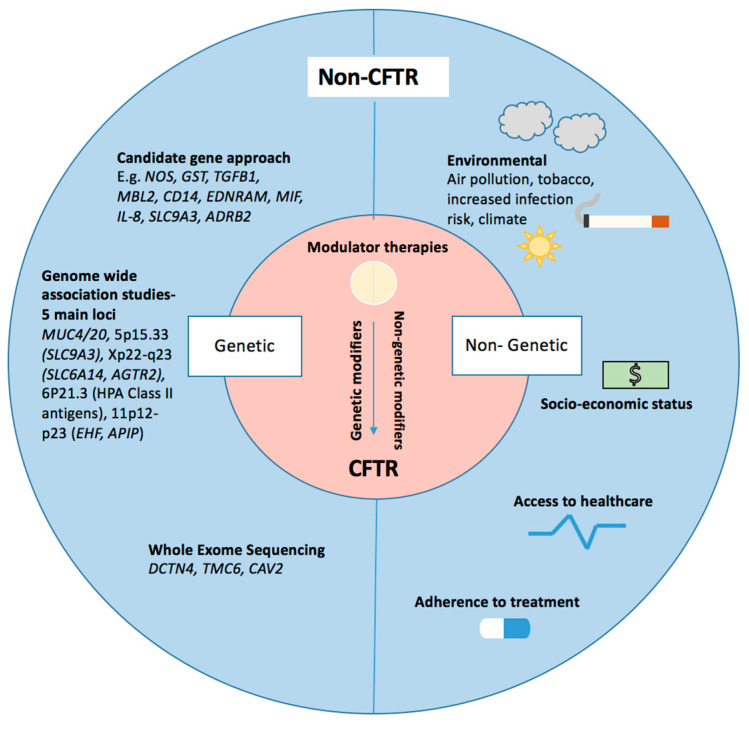
Modifiers of cystic fibrosis (CF) respiratory pathophysiology—key genetic and non-genetic modifiers are illustrated. Genetic modifiers listed have been identified through various techniques including a candidate gene approach, genome-wide approaches and whole exome sequencing. Non-genetic modifiers have been highlighted.

**Table 1 genes-12-00562-t001:** Cystic fibrosis transmembrane conductance regulator (*CFTR*) class mutations and consequent impact on CFTR protein.

Class of Mutation	Impact on CFTR Protein and Example Mutations
I	Defect in protein synthesis, e.g., G542X, R553X, R1162X, W1282X
II	Defect in protein trafficking to cell membrane, e.g., G85E, I507del, F508del, N1303K
III	CFTR protein reaches the cell membrane but defect in channel gating, e.g., S549R, G551D, G1349D
IV	CFTR protein reaches the cell membrane but there is defective conductance, e.g., R117H, R334W, D1152H
V	Normal folding of the CFTR protein with normal function, but amount of protein made is insufficient, e.g., A455E, 2789+5G>A, 3849+10kbC>T
VI	CFTR protein is produced and is in the correct location but has reduced stability with rapid turnover, e.g., F508del, Q1411X

## Data Availability

Not applicable.

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
