# Peer review of "Cystic Fibrosis Lung Disease Modifiers and Their Relevance in the New Era of Precision Medicine"

_genes, 2021, doi:10.3390/genes12040562_

Round 1
Reviewer 1 Report
The authors have written a good review that covers the gambit of gene modifications in CFTR gene or other related genes that might contribute to the emergence of CFTR.
- The major point that is lacking in this review is a discussion of the cells that each of these genetic modulations might be effecting. In the age of single cell resolution studies this needs to be incorporated in a review. For example, CFTR transcript in the airway is shown to be highly expressed in Ionocytes but also broadly expressed in other cells like club cells. What about the other genes discussed in this review – are they functional only in some lung cells? Having that information tremendously improves the significance of this review for the readers.
- An illustration that highlights the CFTR pathway and the role of the other CF disease gene modifiers in that illustration would be helpful.
- Finally a clear separation in mentioning if the results are from human or mouse studies would be helpful
It’s a good effort
Author Response
The major point that is lacking in this review is a discussion of the cells that each of these genetic modulations might be effecting. In the age of single cell resolution studies this needs to be incorporated in a review. For example, CFTR transcript in the airway is shown to be highly expressed in Ionocytes but also broadly expressed in other cells like club cells. What about the other genes discussed in this review – are they functional only in some lung cells? Having that information tremendously improves the significance of this review for the readers.
Many thanks for your review and feedback. Whilst this would also be of potential interest to the reader we did not necessarily feel it was required in this review which aims to provide an up to date overview of CF modifiers, impact on clinical phenotype and consideration of role of CF modifiers in precision medicine.
An illustration that highlights the CFTR pathway and the role of the other CF disease gene modifiers in that illustration would be helpful.
Many thanks for this recommendation. This was considered but wanted focus to be outside CFTR pathway and felt it would be difficult to select just a few of the many genes discussed to illustrate in a diagram and not others. Hence decision to present the gene modifiers collectively in the diagram provided.
Finally a clear separation in mentioning if the results are from human or mouse studies would be helpful
Thank you for your feedback. Some studies already expanded on. Did feel it wouldn't necessarily add to the manuscript to include differentiations on study types where not already done so.
Reviewer 2 Report
Sepahzad, Morris-Rosendahl and Davies choose modifying genes of cystic fibrosis lung disease as a topic for a review and put their work into perspective from (and for) a personal genomics approach.
Undouble, this a laudable subject that has inspired many modifier gene studies which at their core have the intent to use genetic data to define therapeutic targets and to stratify therapeutic management of individual patients.
Initially, candidate gene driven approaches have been launched as small-scale observations from single clinics. One example, considered rightfully by the authors, is the work on ADRB2 and NOS by Grasemann and colleagues. Next, several multicentric studies to identify CF modifying genes have been launched. The European CF twin and sibling study has been done in an association study design on patient pairs with extreme phenotypes. Drumm, Knowles and colleagues have conducted a candidate driven association study on unrelated patients in the USA (and while these were two competing studies, the principle investigators have exchanged many ideas). A twin and sibling study in North America has been conducted at JHU (Cutting and colleagues). Some, but by far not all, of these data have been merged whenever ELSI regulations were compatible. Moreover, the next generation of CF gene modifier studies by Corvol, Strug and colleagues have combined more patient cohorts in larger meta studies with better power to detect meaningful candidate genes.
This small historic excursion shall serve as an explanation to the authors why the reviewer thinks that their reference list is incomplete and that especially their candidate gene list selection is strongly selective. Authors state that “….. many modifier gene studies have been conducted; here we focus on the most significant of these …”, which explains a selection. However, one criterium for significant can be “earlier than the others” or “found in more than one study”, the latter of which implies that findings are robust. Furthermore, some minor mistakes in the data presentation shows that the authors are closer to personal genomics than to cystic fibrosis genetics & genomics. Having stated that, the reviewer apologizes for suggesting further references that the authors might have overlooked (not intentionally, the reviewer is convinced, but citations in a review paper should reflect the field completely).
Chronologically, the following text passages need clarification / editing and/or extension:
- page 2, line 52 to 55 and table 1: assignment of F508del-CFR to its mutation classes does not match between text and table
- page 2, line 59-60 and following lines in this paragraph (“… whilst CFTR mutation class was a reasonable predictor of pancreatic disease, it was poor for the lung [4].”). To this statement, data from Kerem from 1990 is reasonable to quote, however PMID: 11463149 from 2000 (EU data) would complement reference Vanscoy et al from 2007. These three publications agree on “poor genotype-phenotype relationship for the lung”. PMID:11463149 (2000) as well as Vanscoy et al (2007) use twin data to assess the impact of modifying genes on lung function data.
- Figure 1, list of candidate genes. Missing, as it was replicated: IL1B (US: PMID: 19431193; EU: PMID: 21731057) and its receptor IL1R (PMID: 29284683) . Missing large study: PMID: 20837493; this is the data summary of EUCFTSib (J Med Genet 2011). J Med Genet 2011 is not just another modifier gene study – it summarizes data on candidate genes, most of these have been selected based on a transcriptome screen on F508del-CFTR homozygous tissue for differential expression (CF vs on-CF as well as endophenotypes F508del-residual chloride secretion). Thus, this is something of a OMICs-inspired candidate gene study, not a true GWAS, but not “only” a candidate gene study either. Moreover, several CF modifier genes therein have been characterized for their association by NPD, close to “lung disese” and in vivo, thus “close to patients”. This should well be within the scope of this review.
- Figure 1: the reviewer strongly suggests to reference all genes and non-genetic factors that is mentioned in this figure at least once. The reviewer has noticed that references to non-inherited modifiers are given later in the text, however, treatment adherence (while true and likely the most important modifier, albeit difficult to prove) is not substantiated with appropriate public health data yet by the reference list. The authors might consider the following: this is a very well-designed, well-organized and good-to-understand graphic representation of a very complex problem. Readers need to be aware of the basis for the items in this figure, be it common sense (compliance) or scientific data from different studies (IL1R/IL1B).
- Paragraph 4, candidate gene studies: the reviewer feels that the following is missing here: ILB/IL1R pathway, IFNGR1/STAT3 pathway, TNFalpha/TNFR (for TNFR: including NPD association aka F508del-CFTR function in the upper respiratory tract), ENaC as “the most easy to place on a candidate gene list target”, as ENaC is the counterion channel in all CFTR-expressing epithelia – certainly, these are not more or less important than those the authors mention? MUC5 was candidate-based selected & well-proven a reasonable candidate gene for pulmonary function (Knowles, Chapel Hill, and colleagues; US study)
- Paragraph 4, candidate gene studies: AAT (not considered by the authors), TGFB1 and MBL2 share one peculiar feature among candidate gene studies: it has been studied by more than one group and the findings on whether or not it is a modifier differ (AAT, MBL2) or, if agreed on the modifying gene, the risk alleles differ (TGFB1; Arkwright, UK (Leu10 is mild) versus Drumm, US (Pro10 is mild). The current summary reads as if these genes are all well-substantiated by functional data and by association studies.
- Paragraph 4.5, cell surface receptors: Text reads: “Studies have considered the role of toll-like receptors, which are important in host defense, but to date no associations have been found” -> weak association signal for TLR4 in PMID: 20837493. Beyond, CD95 is characterized genetically (PMID: 18685642) and functionally (PMID: 27616356). TNFR, IFNGR1 also would fit this description. While it should not be the goal of the authors to quote all EU data comprehensively, the caption “cell surface receptors” is very well taken and should inform the readers which cell surface receptors are already defined as CF modifiers.
- Paragraph 4.7: “IL8” -> best changed to interleukins? Summarizing findings from IL8, IL1B, and maybe more (reference 16)?
- Page 6, line 227 and ff (on EHF from GWAS): EHF, found by Wright et al (US) has been replicated (PMID: 24105369; EU data).
- Paragraph 7, non-genetic modifiers: several of these items have been recognized as survival determinants - their impact differs by birth year. This might very well explain why Arwkwright (2000, UK) and Drumm (2005, US) do not agree on TGFB1. Impact of survivor effects in CF modifying gene studies have been shown for TGFB1 (PMID: 17406643) and IL1R (PMID: 29284683).
- Page 7, line 257-260, text reads : “ … one of the greatest contributors to health outcomes is adherence to prescribed therapies; studies of behavior have demonstrated that healthcare professionals are poor both at estimating patient adherence and at encouraging it. Numerous apps are being developed specifically to address this unmet need. …”. True, important and valid. However, can this be supported by references?
- Discussion paragraph, section 9, on whether or not modifying genes can assist in a clinically meaningful way. One argument is missing in the reviewer’s opinion: If infectiology-related genes can be targeted, can these be seen as driver for future therapeutic stratification? Inflammation treatment for several interleukins and their receptor-driven pathways are available. The relevance of anti-inflammatory management might also be more relevant with age & disease progression (or: age and disease state prior to onset of CF modulator-corrector-potentiator-therapy). For CF-related diabetes, personal genomics is suggested twice already by polygenic risk scores (most recent: PMID: 31697830)
- Discussion paragraph, page 9, line 362 and surrounding text: agreed, several gene modifier studies are contradictory. But: some modifier genes were independently discovered/replicated in very different settings (competitor studies, different mode of patient selection and different approach to genotyping). See above. This should be explained to the reader, the misinterpretation “the findings don’t match, hence the genes are not important” should be prevented by stating explicitly where studies (from different patient cohorts) agree.
- Same page, line 375 and 376, on what might be the cause for nonagreeable findings between studies: birth year distribution / survivor effect is missing as a major confounder (see above).
Author Response
page 2, line 52 to 55 and table 1: assignment of F508del-CFR to its mutation classes does not match between text and table
Thank you for highlighting this. Text has been edited/removed so that it correlates with table
page 2, line 59-60 and following lines in this paragraph (“… whilst CFTR mutation class was a reasonable predictor of pancreatic disease, it was poor for the lung [4].”). To this statement, data from Kerem from 1990 is reasonable to quote, however PMID: 11463149 from 2000 (EU data) would complement reference Vanscoy et al from 2007. These three publications agree on “poor genotype-phenotype relationship for the lung”. PMID:11463149 (2000) as well as Vanscoy et al (2007) use twin data to assess the impact of modifying genes on lung function data.
Thank you for signposting this- reference now included
Figure 1, list of candidate genes. Missing, as it was replicated: IL1B (US: PMID: 19431193; EU: PMID: 21731057) and its receptor IL1R (PMID: 29284683) . Missing large study: PMID: 20837493; this is the data summary of EUCFTSib (J Med Genet 2011). J Med Genet 2011 is not just another modifier gene study – it summarizes data on candidate genes, most of these have been selected based on a transcriptome screen on F508del-CFTR homozygous tissue for differential expression (CF vs on-CF as well as endophenotypes F508del-residual chloride secretion). Thus, this is something of a OMICs-inspired candidate gene study, not a true GWAS, but not “only” a candidate gene study either. Moreover, several CF modifier genes therein have been characterized for their association by NPD, close to “lung disese” and in vivo, thus “close to patients”. This should well be within the scope of this review.
Section on cytokines expanded. Includes IL1B and receptor IL1R and references highlighted (PMID 19431193 and PMID 29284683
Figure 1: the reviewer strongly suggests to reference all genes and non-genetic factors that is mentioned in this figure at least once. The reviewer has noticed that references to non-inherited modifiers are given later in the text, however, treatment adherence (while true and likely the most important modifier, albeit difficult to prove) is not substantiated with appropriate public health data yet by the reference list. The authors might consider the following: this is a very well-designed, well-organized and good-to-understand graphic representation of a very complex problem. Readers need to be aware of the basis for the items in this figure, be it common sense (compliance) or scientific data from different studies (IL1R/IL1B).
Thank you for the feedback on the figure and highlighting the need for further references to substantiate the point on treatment adherence. As recommended each genetic and non-genetic factor is referenced in the text and I have added several references to evidence the impact of treatment adherence.
Paragraph 4, candidate gene studies: the reviewer feels that the following is missing here: ILB/IL1R pathway, IFNGR1/STAT3 pathway, TNFalpha/TNFR (for TNFR: including NPD association aka F508del-CFTR function in the upper respiratory tract), ENaC as “the most easy to place on a candidate gene list target”, as ENaC is the counterion channel in all CFTR-expressing epithelia – certainly, these are not more or less important than those the authors mention? MUC5 was candidate-based selected & well-proven a reasonable candidate gene for pulmonary function (Knowles, Chapel Hill, and colleagues; US study)
Thank you. I have included data evidencing the role of ILB/ILR pathway. Section 4.8 changed to ion channels to include SLC9A3 and ENaC. I have also included data on MUC5 (new section 4.9).
Paragraph 4, candidate gene studies: AAT (not considered by the authors), TGFB1 and MBL2 share one peculiar feature among candidate gene studies: it has been studied by more than one group and the findings on whether or not it is a modifier differ (AAT, MBL2) or, if agreed on the modifying gene, the risk alleles differ (TGFB1; Arkwright, UK (Leu10 is mild) versus Drumm, US (Pro10 is mild). The current summary reads as if these genes are all well-substantiated by functional data and by association studies.
Thank you for this point. I have extended the section on MBL2 to provide a more balanced view and opposing findings. I have also amended the section of TGFB1 to include different findings in risk alleles.
Paragraph 4.5, cell surface receptors: Text reads: “Studies have considered the role of toll-like receptors, which are important in host defense, but to date no associations have been found” -> weak association signal for TLR4 in PMID: 20837493. Beyond, CD95 is characterized genetically (PMID: 18685642) and functionally (PMID: 27616356). TNFR, IFNGR1 also would fit this description. While it should not be the goal of the authors to quote all EU data comprehensively, the caption “cell surface receptors” is very well taken and should inform the readers which cell surface receptors are already defined as CF modifiers.
Amended to include data for weak signal for TLR4.
Paragraph 4.7: “IL8” -> best changed to interleukins? Summarizing findings from IL8, IL1B, and maybe more (reference 16)?
Thank you- Agree and amended and now including IL1B.
Page 6, line 227 and ff (on EHF from GWAS): EHF, found by Wright et al (US) has been replicated (PMID: 24105369; EU data).
Thank you for highlighting this. Reference has been added
Paragraph 7, non-genetic modifiers: several of these items have been recognized as survival determinants - their impact differs by birth year. This might very well explain why Arwkwright (2000, UK) and Drumm (2005, US) do not agree on TGFB1. Impact of survivor effects in CF modifying gene studies have been shown for TGFB1 (PMID: 17406643) and IL1R (PMID: 29284683).
Thank you for flagging this important bias. I have made reference to this in the text and used the reference kindly provided
Page 7, line 257-260, text reads : “ … one of the greatest contributors to health outcomes is adherence to prescribed therapies; studies of behavior have demonstrated that healthcare professionals are poor both at estimating patient adherence and at encouraging it. Numerous apps are being developed specifically to address this unmet need. …”. True, important and valid. However, can this be supported by references?
Many thanks- References now added to support these points
Discussion paragraph, section 9, on whether or not modifying genes can assist in a clinically meaningful way. One argument is missing in the reviewer’s opinion: If infectiology-related genes can be targeted, can these be seen as driver for future therapeutic stratification? Inflammation treatment for several interleukins and their receptor-driven pathways are available. The relevance of anti-inflammatory management might also be more relevant with age & disease progression (or: age and disease state prior to onset of CF modulator-corrector-potentiator-therapy). For CF-related diabetes, personal genomics is suggested twice already by polygenic risk scores (most recent: PMID: 31697830)
Many thanks for highlighting the role of modifiers in a clinical determination and PRS studies, we have included this in the second last paragraph of the discussion, where we felt it best fit.
Discussion paragraph, page 9, line 362 and surrounding text: agreed, several gene modifier studies are contradictory. But: some modifier genes were independently discovered/replicated in very different settings (competitor studies, different mode of patient selection and different approach to genotyping). See above. This should be explained to the reader, the misinterpretation “the findings don’t match, hence the genes are not important” should be prevented by stating explicitly where studies (from different patient cohorts) agree. Same page, line 375 and 376, on what might be the cause for nonagreeable findings between studies: birth year distribution / survivor effect is missing as a major confounder (see above).
We have read the paper through, and do not consider that we’ve suggested data from conflicting studies renders a particular gene unimportant. In the interests of maintaining brevity, we have not therefore further expanded upon this.
Round 2
Reviewer 1 Report
Accept as is
Author Response
Many thanks again for your feedback of this review
Reviewer 2 Report
Sepahzad, Morris-Rosendahl and Davies have provided a revised manuscript reviewing modifying genes of cystic fibrosis lung disease. the reviewer thanks the authors for their effort in improving the manuscript considerably, making it more informative for the reader. As for the previously submitted manuscript, an unbalanced citation has been noticed, the focus of the reviewer is now on references. As always for reviews, the list of references is long – still, the following items have been overlooked by the authors as the text does not list a reference in the pdf provided by The Journal's webpage:
page 1, line 36 to 37:” In the lungs this leads to bacterial infection, inflammation, airway scarring (bronchiectasis) and early mortality from respiratory failure.” -> true and well-known, but needs a reference, likely to a recent review
page 1, lines 37 to 39:” Other complications include pancreatic exocrine insufficiency, bowel disease (meconium ileus in newborns and distal intestinal obstruction syndrome (DIOS) in older people (2)), diabetes and liver disease.” -> is reference 2 intended to support all claims or restricted to age dependency of MI/DIOS?
page 1, line 39 to 41: “CFTR 39 dysfunction in the genitourinary tract leads to congenital bilateral absence of the vas deferens (CBAVD) and male infertility.” -> true and well-known, but needs reference, likely to a recent review
page 1, line 41 to 43: “Initially considered a paediatric disease with poor 41 prognosis, CF is a very different disease today; over 50% of patients are now aged over 18 years in many regions.” In contrast to the previous statements (which will not raise a debate among CF caregivers), this might be the subject of debate, depending on the CF center that you work with. Please provide the reference that you base your estimate on.
page 5, line 164 to 166 "A different study found that the informative microsatellite marker within intron 1 of IL1R detects a survival advantage for patients with CF and supports the potential role of interleukin 1 receptor (IL1R) in the pathogenesis of CF." -> missing reference
page 5, line 181 to 182: “a potassium sparing ENaC antagonist” To the best of the reviewer’s knowledge, amiloride directly inhibits ENaC. What exactly do the authors mean by potassium-sparing? Please reword to reflect current knowledge on the mode of action of amiloride.
Page 5, line 182 to 183:” Amiloride, a potassium sparing ENaC antagonist was trialled initially with some positive effect in reducing mortality rates. Its use was terminated however due to the risk of hyperkalemia.” -> missing reference
Page 6 line 214 to 215: “There have been three major GWAS which have identified a number of variants found to have associations with differing phenotypes seen in CF.” Please provide the three respective references here as a support.
Page 6 line 223 to 224:” Together they play a crucial role in providing an osmotic barrier and contribute to the periciliary brush layer.” -> missing reference
Page 7, line 238 “No further publications in this area are available.” -> Incorrect. While I am uncertain why the work of Pereira and colleagues has not been considered by the authors, these two publications on SLC9A3 have been easy to retrieve via Pubmed (genotype-phenotype studies: 2018: PMID: 29635781; 2017: PMID: 28756021; many further functional studies are available on SLC9A3 and cystic fibrosis). The reviewer strongly suggest to remove sentences that claim that “no further information is available”, as even if these should be correct right now, they might turn out to be outdated soon (other example: “There are no further studies specifically investigating AGTR2.”, page 7 line 243).
Page 7, line 257 to 261: “This induced increase in CFTR function was also observed to 257 lead to an increase in the ASL height. They also found that potentiation of F508del-CFTR channel function in CF cells induced by SLC6A14 arginine uptake occurred via the nitric oxide (NO) signaling pathway. Intriguingly, they suggest that SLC6A14 activation may be considered as a complement therapy to CFTR correction and potentiation in CF patients.” All refer to reference 59? Please specify.
Discussion paragraph, page 10: While a discussion paragraph also serves to convey the author’s assessment (with which the reviewer generally agrees), some statements might not withstand the test of time as the method on how big data can be evaluated and integrated into existing (and growing) knowledge databases will improve to extract valuable information from resources that cannot be fully interpreted today. To ensure that this paragraph withstands the test of time, the following statements should to be substantiated with a reference (that will in the future show that by 2021, the assessment was state-of-the-art):
line 422 to 426: “There are also challenges when an identified locus is associated with disease in one ethnic/ regional group and not another as described above. This may imply differences in risk allele frequency between different geographical regions – hence much larger sample sizes are needed to detect the association.” -> please provide at least one example from the general literature (or a review discussing this topic in detail)
line 439 to 440: “To date, the only genetic modifier reported is SLC26A9, a poorly understood solute carrier thought to interact with cell surface CFTR.” And line 441 to 442: “Two studies reporting associations as well as laboratory data support the association, although population differences mean that this effect is currently not consistent and further work is needed.” The authors appear to have concluded that once quoted previously in this manuscript, as second reference is unnecessary in the discussion. However, this is a recent and novel finding that deserves in the reviewer’s opinion a second easy-to-retrieve keys reference in the discussion section. Also, the molecular mechanism remains unresolved (as rightfully stated), hence, the authors might want to make sure that their statement is anchored to the literature that is currently available.
Author Response
page 1, line 36 to 37:” In the lungs this leads to bacterial infection, inflammation, airway scarring (bronchiectasis) and early mortality from respiratory failure.” -> true and well-known, but needs a reference, likely to a recent review
Many thanks. Two new references added.
page 1, lines 37 to 39:” Other complications include pancreatic exocrine insufficiency, bowel disease (meconium ileus in newborns and distal intestinal obstruction syndrome (DIOS) in older people (2)), diabetes and liver disease.” -> is reference 2 intended to support all claims or restricted to age dependency of MI/DIOS?
Thanks for this- Reference 2 is. intended to support age dependency of MI/DIOS. Sentence has been re-worded and new reference added to support other features.
page 1, line 39 to 41: “CFTR 39 dysfunction in the genitourinary tract leads to congenital bilateral absence of the vas deferens (CBAVD) and male infertility.” -> true and well-known, but needs reference, likely to a recent review
Reference added
page 1, line 41 to 43: “Initially considered a paediatric disease with poor 41 prognosis, CF is a very different disease today; over 50% of patients are now aged over 18 years in many regions.” In contrast to the previous statements (which will not raise a debate among CF caregivers), this might be the subject of debate, depending on the CF center that you work with. Please provide the reference that you base your estimate on.
Many thanks for your feedback. Sentence re-written and reference now given
page 5, line 164 to 166 "A different study found that the informative microsatellite marker within intron 1 of IL1R detects a survival advantage for patients with CF and supports the potential role of interleukin 1 receptor (IL1R) in the pathogenesis of CF." -> missing reference
Many thanks for flagging this. A reference has now been added.
page 5, line 181 to 182: “a potassium sparing ENaC antagonist” To the best of the reviewer’s knowledge, amiloride directly inhibits ENaC. What exactly do the authors mean by potassium-sparing? Please reword to reflect current knowledge on the mode of action of amiloride.
Reworded
Page 5, line 182 to 183:” Amiloride, a potassium sparing ENaC antagonist was trialled initially with some positive effect in reducing mortality rates. Its use was terminated however due to the risk of hyperkalemia.” -> missing reference
Many thanks- reference now added
Page 6 line 214 to 215: “There have been three major GWAS which have identified a number of variants found to have associations with differing phenotypes seen in CF.” Please provide the three respective references here as a support.
Three references now added
Page 6 line 223 to 224:” Together they play a crucial role in providing an osmotic barrier and contribute to the periciliary brush layer.” -> missing reference
Thanks- reference added
Page 7, line 238 “No further publications in this area are available.” -> Incorrect. While I am uncertain why the work of Pereira and colleagues has not been considered by the authors, these two publications on SLC9A3 have been easy to retrieve via Pubmed (genotype-phenotype studies: 2018: PMID: 29635781; 2017: PMID: 28756021; many further functional studies are available on SLC9A3 and cystic fibrosis). The reviewer strongly suggest to remove sentences that claim that “no further information is available”, as even if these should be correct right now, they might turn out to be outdated soon (other example: “There are no further studies specifically investigating AGTR2.”, page 7 line 243).
Line stating “no further information available has now been removed”.
Page 7, line 257 to 261: “This induced increase in CFTR function was also observed to 257 lead to an increase in the ASL height. They also found that potentiation of F508del-CFTR channel function in CF cells induced by SLC6A14 arginine uptake occurred via the nitric oxide (NO) signaling pathway. Intriguingly, they suggest that SLC6A14 activation may be considered as a complement therapy to CFTR correction and potentiation in CF patients.” All refer to reference 59? Please specify.
Discussion paragraph, page 10: While a discussion paragraph also serves to convey the author’s assessment (with which the reviewer generally agrees), some statements might not withstand the test of time as the method on how big data can be evaluated and integrated into existing (and growing) knowledge databases will improve to extract valuable information from resources that cannot be fully interpreted today. To ensure that this paragraph withstands the test of time, the following statements should to be substantiated with a reference (that will in the future show that by 2021, the assessment was state-of-the-art):
Many thanks for this point which was considered but did not feel it was necessary to include a further reference
line 422 to 426: “There are also challenges when an identified locus is associated with disease in one ethnic/ regional group and not another as described above. This may imply differences in risk allele frequency between different geographical regions – hence much larger sample sizes are needed to detect the association.” -> please provide at least one example from the general literature (or a review discussing this topic in detail)
Many thanks- Reference added
line 439 to 440: “To date, the only genetic modifier reported is SLC26A9, a poorly understood solute carrier thought to interact with cell surface CFTR.” And line 441 to 442: “Two studies reporting associations as well as laboratory data support the association, although population differences mean that this effect is currently not consistent and further work is needed.” The authors appear to have concluded that once quoted previously in this manuscript, as second reference is unnecessary in the discussion. However, this is a recent and novel finding that deserves in the reviewer’s opinion a second easy-to-retrieve keys reference in the discussion section. Also, the molecular mechanism remains unresolved (as rightfully stated), hence, the authors might want to make sure that their statement is anchored to the literature that is currently available.
Reference provided again as advised
Thank you for the time spent reviewing thee manuscript and for the valuable suggestions.